# Heritage, Tourism and Local Development in Peripheral Rural Spaces: Mértola (Baixo Alentejo, Portugal)

**F. Javier García-Delgado [1], Antonio Martínez-Puche [2] and Rubén C. Lois-González [3,*]**

1 Department of History, Geography and Anthropology, University of Huelva, 21071 Huelva, Spain; fcogarci@uhu.es
2 Department of Human Geography, University of Alicante, San Vicente del Raspeig, 03080 Alicante, Spain; antonio.martinez@gcloud.ua.es
3 Department of Geography, University of Santiago de Compostela, Santiago de Compostela, 15703 A Coruña, Spain
* Correspondence: rubencamilo.lois@usc.es

**Abstract:** In the context of multiple repurposing of rural spaces, tourism represents a path for development, with the potential to revitalize these areas. The conservation and restoration of heritage, and its promotion through tourism, can become an opportunity for local development, in which a range of stakeholders fulfil different roles in the carrying out of the processes involved. The aim of the study was to analyse the heritagisation processes and their tourist value enhancement and how it affects local development in Mértola (Baixo Alentejo, Portugal). A series of interviews with the chief stakeholders in the process were conducted, from which the contexts and conceptualisations of development were determined. On the basis of secondary data in terms of statistics, an analysis of the impacts of the process of heritagisation and the development of tourism was undertaken. The main conclusions drawn by the research are the following: (a) the importance of the process of heritagisation in Mértola; (b) the viability of the project, given the cost and lack of comprehensive conservation, in creating a unified whole; (c) the performance of, and power relationships between, the various stakeholders; (d) the limited participation of locals due to disaffection with the project; (e) the correlation between heritage, rural tourism, and local development.

**Keywords:** peripheral areas; local development; heritagisation; sustainable rural tourism; stakeholders; disaffected citizens

## 1. Introduction

Rural areas have often experienced a deepening crisis as a result of the effects of globalisation, economic cycles, new production practices, and sociological and cultural changes, all of which have forced local development to adopt multifunctional approaches [1,2] and economic diversification [3]. These processes have been comparably common in "lagging rural regions" [4] (p. 347), which characteristically lack the critical mass to be able to compete and suffer from the decline of traditional activities [4] and a marked peripheralisation [5]. Consequently, growth within these communities is highly dependent on their capacity for adaptive [2] and innovative [6] strategies, which can allow its development, overcoming the centre–periphery models [4].

By these means, rural areas can become "locations for the stimulation of new socio-economic activity" [4] (p. 347) through diversification [1,4]: leisure, rural tourism, catering establishments, biodiversity conservation, housing expansion, and utilization of the natural and cultural heritage.

Also, the reinterpretation of other traditional uses of heritage such as agriculture, agribusiness, crafts, and quality products, among others, is possible [4,7].

Two perspectives of the rural landscape and its resources have emerged [8]: (a) the external view: as a recreational space which needs to be regulated if it is to be preserved and enjoyed; (b) the internal view: as a habitable space, the legislation concerning which acts as a barrier to the everyday activities and practices of the population. Residents tend to take a utilitarian and pragmatic perspective view of rural spaces based on production, often at odds with environmental regulation of its natural resources, while visitors and tourists tend towards a more aesthetic or consumer-centred perspective, which favours legislating its uses [9]. In this regard, the EU policies have articulated guidelines for the diversification and improvement of agricultural production, the prevention of rural depopulation, and the generation of employment and income. Rural communities have thus seen their social and cultural capital become their main heritage asset [10,11].

Rural areas on the periphery base their strategies for development on traditional activities [4]. In such a context, tourism becomes a challenge [12] and takes on a dual role as: (a) an agent of diversification and regeneration of the traditional way of life [13]; and (b) a means of strengthening the processes of local development [14]. Control over these processes on the part of local residents enables them to ensure that this development is both sustainable and beneficial [15]. Nevertheless, it is possible for tourism to be overvalued as a panacea for the decline in rural conditions [12,16], as the political and popular discourses testify, and for an area's limitations in terms of development to be pushed to the background [17], while its resources and potential are foregrounded [18]. The fact that not all locations are equally open to the development of tourism, enjoy the same degree of popularity, or have the same advantages is often forgotten, thus fuelling the contrary viewpoint that regards transformation in the name of tourism as a commodification of the rural environment [19].

To this can be added an additional layer of complexity with regards to studies into rural tourism. There is currently a wide variety of models, activities, and types of accommodation, which in turn are often in need of a "new generation" of rural tourism, based on the management of smart, virtually oriented destinations [20,21], a deeper understanding of the market, and fully integrated professional management systems oriented towards sustainability [22]. The phrase "rural tourism" is frequently employed as an umbrella term defined by geographical location, whereby activities coming within its scope have nothing in common beyond the fact that they take place in a rural context, as opposed to an urban one [23]. Although rural tourism places a premium on existing heritage to create value, it is nevertheless a rapidly evolving area, with significant challenges and business opportunities [24]. Consequently, theoretical accounts and policy decisions highlight the importance of a grassroots approach to rural development, the active involvement of the local community, and the development of small-scale projects underlining "tradition, character and culture" [25] (p. 108). Also fundamental to improving the perspectives of the sector is the involvement of local political leaders in mapping out processes, putting essential services in place, and improving the business environment.

Indeed, in peripheral areas, which have seen a decline in traditional activities [12] and where opportunities are scarce, "any economic diversification is likely to be welcomed" [15] (p. 532) and "tourism is a desirable diversifier" [11] (p. 391). For the "boring peripheries" and in-between areas [26] (p. 740), tourism represents a new means of regional development [27], although it is yet to be seen whether the equation when tourism equals development is more than wishful thinking [12,18]. Much will depend on the local and temporal context, the political will, the cultural and socioeconomic resources available in the territory, and stakeholder commitment. All the foregoing aspects will be dealt with in this case study, in which, additionally, a dialectic will be established between the 'heritagisation' and the exploitation of heritage for the purposes of tourism.

Tourism activities in the periphery can be a viable option for achieving economic development as an effective source of income and employment [12,28], tackling the issues of access whilst rejuvenating and retaining the population [6,18,29]. In this respect, tourism is often regarded as a "catalyst for innovative local development" [11] (p. 383), enabling the reduction of regional disparities [18]. Nonetheless, regions may not always obtain better results, despite receiving more funds [18], as these may be poorly managed [30]. Provided tourist attractions are generated in peripheral areas, unique destinations

and products can be consolidated, encouraging where visitors can travel, with the motivation to participate in diverse experiences [29]. Often these attractions are not sufficient to establish an extensive tourism offer and, thus, local development based on tourism [29] since the scale of attraction, the conservation, and the uniqueness factor of the resource is the one that potentially generates growth of other types of tourism and maintains its viability [6,31]. Thus, the degree of peripherality determines the tourist flow, distinguishing between peripheral disconnected destinations [32] and intermediate destinations [26], accessible by road [29], which often receive so-called "autonomous tourism" or "rubber tire traffic" [33], making it possible to generate further development in destinations closer to densely populated areas [22], although, consequently, it might result in overexploitation and fragility of the spaces [34,35]. The preservation, intervention, and recuperation of heritage, and the value this brings to an area, become an opportunity for sustainable local development [36], contributing through tourism projects that seek to "design new spaces" [37] (p. 290), in which different stakeholders take part. The process of "heritagisation" focuses on those elements that are unique to a particular rural area, rooted in its history, and identifiable as a "marker of regional identity" [37] (p. 275). However, the sheer range and scale of heritage makes it difficult to conserve and promote, particularly if the economic resources are limited [38]. Nor is this aided by the confusion between the notions of resource and product (the latter meeting demand and having a price) [37]. The process by which heritage resources in rural areas are converted into tourism products needs to be located in a post-Fordist context [39]. It is a process which, since the early 1990s and as recognised by ICOMOS [40], has witnessed an expansion into the cultural space [41].

The conjunction of cultural heritage and tourism has been widely studied [41–43], as it opens up possibilities for the economic development of places with a depth of heritage, although at the same time it creates challenges for the management of attractions [43,44]. Such is its importance that it is institutionalised in public policy and local development [45], creating an interdependence between heritage conservation and the development of tourism [42], although this relationship is not without contradictions and conflicts [43]. In this manner, both positive and negative effects derive from the conjunction, most of which are common to rural and heritage tourism.

Integrated rural tourism (IRT) is an approach that seeks to avoid, or at least to mitigate, the problems associated with tourism in rural areas. In this endogenous model, local actors are important because they "benefit from policies that empower them and enhance their long-term well-being" [4] (p. 363). By contrast, cultural tourism is promoted as a means of economic and social diversification [29]. However, in terms of the institutional context, the management of heritage differs from those organisations that regard resources more as assets for tourism [42]. The emphasis is on protecting and preserving heritage rather than ensuring that it returns a profit [43] (p. 33). In order to satisfy advocates of these opposing perspectives, it is necessary to investigate points of contact between them [43].

Of fundamental importance to planning tourism is the coordination and collaboration between stakeholders [46–48], essential to which is the collaborative focus at all levels between those responsible for managing heritage and all that goes with it, and those responsible for tourism and all its resources [43]. All interested parties should be involved in the process [49], as success depends on their commitment. Further, according to community participation theory, the inclusion of local residents in the decision-making process is also important [43], as their involvement in the development of projects has a significant impact [46,50]. There is, too, the issue of leadership and the delegation of responsibilities among the stakeholders [47] in determining the social relationships underlying the construction of a tourist territory [51]. The relationship between management and sustainable tourism should also be taken into account [52].

In this regard, the case study of the town of Mértola (Baixo Alentejo, Portugal) is particularly relevant as it embodies the elements and processes discussed above. It is a small town with a population of around 6000, located in a rural area, which has been in demographic and economic decline since the middle of the 20th century due to the loss of traditional primary activities (see Section 2.2). In 1978,

a process of markedly ideological heritagisation was initiated to stimulate local development, which was supplanted at the beginning of the 21st century by a project to expand tourism.

The main objective of this paper was to carry out a diachronic study into the processes involved in heritagisation, from a tourism and local development perspective, and to undertake an analysis of their social, political, and institutional contexts [53]. The study focused on a singular location in the rural periphery, which has been overlaid, like a palimpsest, with an archaeological and material conception of heritage, foregrounding local resources as elements of identity and awareness of the past. Given the need to seek for the alternatives to tackle the structural crisis, a process of tourism valorisation was chosen in the least touristic area of Portugal. Therefore, these processes in the rural context were analysed. Analysis of the processes involved in this shift to rural tourism includes the roles and background of the stakeholders; the measures, instruments, and actions implemented in the course of heritagisation and implantation of cultural tourism; and a critical assessment of the successes, failures, results, and overall impact. Consideration was also given to the lessons that could be drawn from the Museum Town of Mértola project, and which can be transferred to other locations with significant cultural heritage and committed involvement of the stakeholders.

## 2. Methodology and Case Study

### 2.1. Data and Methods

Studying local development through the complex relationships between heritage/tourism and the stakeholders is best achieved by use of a case study approach, by means of collecting in-depth data from a variety of sources [54]. The paradigm has been widely applied to studies of tourism [55], in particular the processes and management of heritagisation [56], roles and relationships between stakeholders and governance [11,28,43,57,58], local/rural development and tourism [25,59], and tourism in the periphery [18,29].

The methodology employed was qualitative, based on interviews intended to collate different opinions and perceptions from the principal actors [57]; to identify social networks and respective power structures [60]; and to determine the effects of heritagisation, the foregrounding of tourism, and the problems deriving from these processes [29,57]. The interviews were semi-structured [61], consisting of open questions, which allowed for digression into related topics of interest [62], enabling us to obtain information on different topics (Table 1). In total five interviews were conducted with actors involved in the processes of heritagisation and the promotion of tourism:

- Interview 1 (hereinafter Int1): political representative of the Town Council, vice president of the Mértola Municipal Chamber (hereinafter CMM).
- Interview 2 (hereinafter Int2): museology and heritage specialist for the CMM; also a member of the Mértola Archaeological Site (hereinafter CAM).
- Interview 3 (hereinafter Int3): archaeologist, director and the founder of the CAM.
- Interview 4 (hereinafter Int4): archaeologist, co-director of the CAM, responsible for management of CAM.
- Interview 5 (hereinafter Int5): local business woman, representative of the tourism sector.

The data obtained from the interviews were complemented by intensive territorial reconnaissance (valuation of the heritage environment, accessibility study, informal interviews with local business interests and residents) and secondary sources centred on: (a) heritage characterisation, the heritagisation process, and tourism promotion in Mértola, based both on the published sources and the planning documents); (b) the prevalent discourses in the conjunction of heritage and tourism, both in published and unpublished research on Mértola; (c) official statistical information for analysing results (Statistics Portugal, hereinafter INE) and official databases (National Tourism Register, hereinafter RNT) [63].

**Table 1.** Questions of the interview.

| Block of Questions | Questions | Obtained Information | | | | | | |
|---|---|---|---|---|---|---|---|---|
| | | t1 | t2 | t3 | t4 | t5 | t6 | t7 |
| All interviewees | Heritage, tourism, or development idea | ▓ | ▓ | | | | | ▓ |
| | Tourism management, organisation, and planning | | | | | ▓ | | |
| | Project objectives with the reference to the Mértola Vila Museu Project | | | | | | | |
| | Management of instruments and tools | ▓ | | | | | | |
| | Project funding | | | ▓ | | | | |
| | Participating actors and/or characteristics | | ▓ | | ▓ | | | |
| | Role of local population | | ▓ | | | | | ▓ |
| | Valuation of resources | | | | | | | ▓ |
| | Heritage, tourism, or development diagnosis | | | | | ▓ | | ▓ |
| | Cooperation, participation, and competition | ▓ | | | | | | |
| | Models consulted | | | | | | | ▓ |
| | Promotion strategies | | ▓ | | | | | |
| | Proposals | | ▓ | | | | | ▓ |
| For companies | Company data, characteristics, and seniority | | | ▓ | | | | |
| | Training, both of employers and employees | | | | ▓ | ▓ | | |
| | Origin of a company | | | ▓ | | | | |
| | Employment generated | | | ▓ | ▓ | | | |

Topics of information: t1: processes; t2: heritage and/or tourism relationships; t3: implementation of initiatives, measures, instruments, and actions; t4: actors involved; t5: objectives; t6: results and/or impacts; t7: sustainable tourism and/or local development.

## 2.2. Case Study: Mértola

The municipality of Mértola is located in the SW of the Iberian Peninsula, in the Beja district (Baixo Alentejo province) of Portugal (Figure 1). It is the sixth largest municipality in Portugal, at 1293 km$^2$, and is divided into 7 smaller areas or "freguesías", considered as parishes (Figure 1).

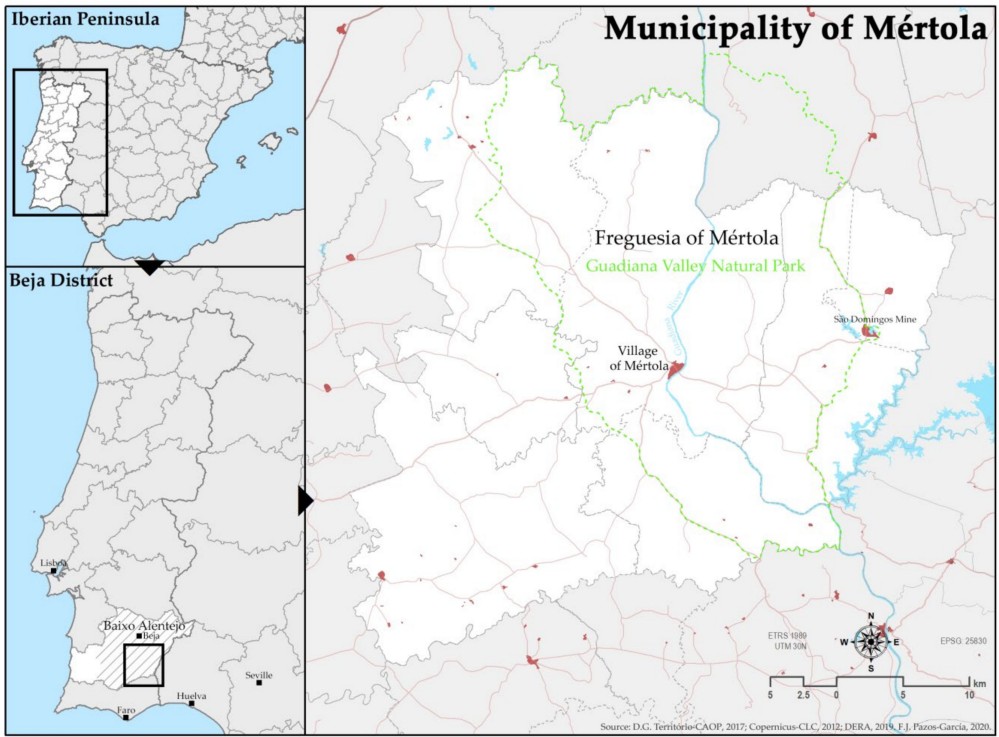

**Figure 1.** Area of the study, location, and the administrative structure.

It is a peripheral area, whose borderland status has caused the crisis to be keenly felt [64]. Due to the fact that the land is unsuitable for arable farming, the main traditional activities have been forestry, animal husbandry, and hunting, distributed among large private estates. In 1995, the Guadiana Valley Natural Park (hereinafter PNVG), covering 47.39% of the municipality, was created to protect its outstanding natural beauty and ecological wealth (Figure 1).

Closure of the mines and the agricultural crisis in the mid-20th century precipitated a period of decline and rural exodus. In 2018, there were 6202 residents, a loss of 76.17% of the 1960 total. It is also an aging population (58.59% ≥ 65), with a very low demographic density (4.80 inhabitants/km$^2$) [65] dispersed over 98 population nuclei [66].

Mértola is equidistant (120 km) from the towns of Faro (Portugal) and Huelva (Spain), and likewise from the major cities of Lisbon and Seville (220 km). The nearest sizeable town is Beja (53 km) (Figure 1). Increased road connectivity from the mid-20th century onwards caused the demise of river transport, relegating the town even further to the backwaters, although improved access to the Algarve and Spain at the beginning of the 21st century went some way to counteract this.

The Alentejo is the least visited region of Portugal [30], especially Baixo Alentejo (Table 2), which has seen very little investment. In spite of this, there are several attractions in Mértola worthy of tourist interest, in the form of cultural heritage, e.g., "vila" of Mértola as the museum town; the natural environment (PNVG); and industrial heritage, such as the São Domingos Mine, a disused open-cast 'Victorian' copper mine on the western fringes of the Iberian Pyrite Belt (Figure 1).

**Table 2.** Tourism importance in Portugal, Baixo Alentejo, and Alentejo (2018).

| Territorial Scope | Guests (Total) | Guests (% of the National Total) | Lodging Capacity (Total) | Lodging Capacity (% of the National Total) |
|---|---|---|---|---|
| Baixo Alentejo | 202,534 | 0.80 | 3010 | 0.71 |
| Alentejo | 1,470,950 | 5.83 | 23,852 | 5.64 |
| Portugal | 25,249,904 | 100.00 | 423,152 | 100.00 |

Source [67].

The town of Mértola itself is a walled hilltop city on the right bank of the Guadiana River (Figure 2), the choice of location being determined by its navigability, namely at 72 km from the river mouth, defensibility, abundance of water, and polymetallic deposits [68]. Within the walls, the town is today typical of modern Portuguese architectural style over an Islamic stratum [69] of considerable historic and aesthetic interest [69]. This heritage began to be valued at the end of the 1970s in the form of the "Mértola Museum Town" project, and since the start of the new millennium, the tourism dimension has been foregrounded. In 2017, Mértola was added to Portugal's Tentative List (TL) (This should not be confused with UNESCO's "World Heritage List" (hereinafter WHL) of sites with World Heritage status. The "Tentative List" is the result of UNESCO's recommendation for member States to "submit their Tentative Lists, properties which they consider to be cultural and/or natural heritage of outstanding universal value and therefore suitable for inscription on the World Heritage List." [70]. Inclusion on the Tentative List is a prerequisite for being declared a World Heritage Site, but does not guarantee inscription on the WHL.) for inclusion on the World Heritage List.

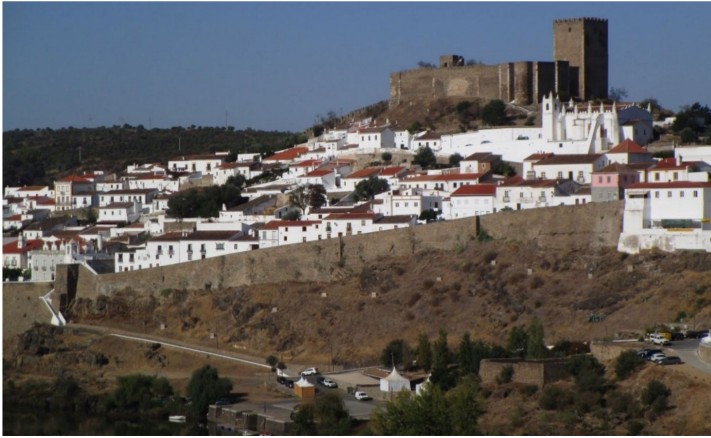

**Figure 2.** View of the town of Mértola from the left bank of the Guadiana river. The wall, the Mother Church, and the castle can be observed.

## 3. Results and Discussion

### 3.1. Actions Towards the Protection, Heritagisation and Enhancement of Tourism

Three buildings in Mértola have been declared national monuments: the church 'Igreja Matriz' (Almohad mosque (12th century), constructed on an early Christian church (6th century), and consecrated after the Reconquista (13th century) [71]) and the 'Torre del Rio'(ancient wharf and fortified port structure (5th century), unique in Portugal, controlling access to the port and the movement of goods [69]) (misleadingly known as the 'Old Bridge' in English, though it is neither a tower nor a bridge), both in 1910, and the 'Castelo de Mértola'(the Muslim fortress (12th century) was remodelled after the Christian Reconquista (13th century) [68]) in 1951 [72] (Figure 3). These declarations of assets have not generated interventions or led to a plan to protect these to be set [73].

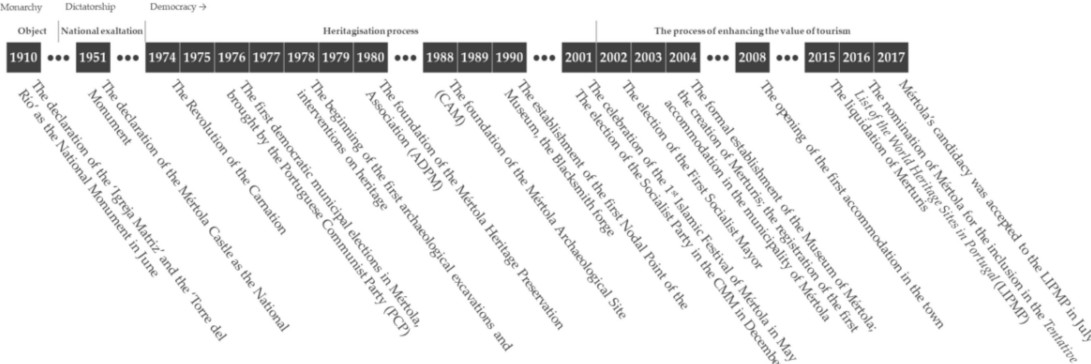

**Figure 3.** The chronology of the main milestones, processes, and political context of the heritagisation and its value for tourism.

The first democratic municipal elections in Mértola (1976) brought the Portuguese Communist Party (hereinafter PCP) (The party continues to take a role in local coalitions to this day.) to power, and the new Mayor set about recovering the town's historical, cultural, and natural heritage [74], with the guidance of researchers from the University of Lisbon [75].

In 1980, the not-for-profit "Mértola Heritage Preservation Association" (hereinafter ADPM) was established with the aim of conserving and promoting the town's heritage [76]. Facing the need to invest in infrastructure and services, the municipal authority delegated this role to the ADPM, both parties sharing political and ideological affinities [75]. In 1988, projects being undertaken nationally were required by the ADPM to be split in two, resulting in the "Mértola Archaeological Site" (CAM) being formed to deal with the material cultural heritage, while the ADPM took responsibility for the natural and ethnographic heritage.

The specialist scientific support supplied by CAM to the heritagisation strategy was channelled through the "Mértola Museum Town Project" (hereinafter PMVM) [73]. According to this plan, the 'vila', to give the town its historical appellation, was conceived of as an open-air museum [76], incorporating a wealth of archaeological and architectural elements into a route around the centre [73]. In this way the entirety of the town was deemed a single resource [77] (p. 236), gradually incorporating new elements, such as the nodal points of the museum, into the whole (Figure 4). Wherever possible, these nodal points are housed in restored buildings [77] at the site of the archaeological finds [75] (Figure 5).

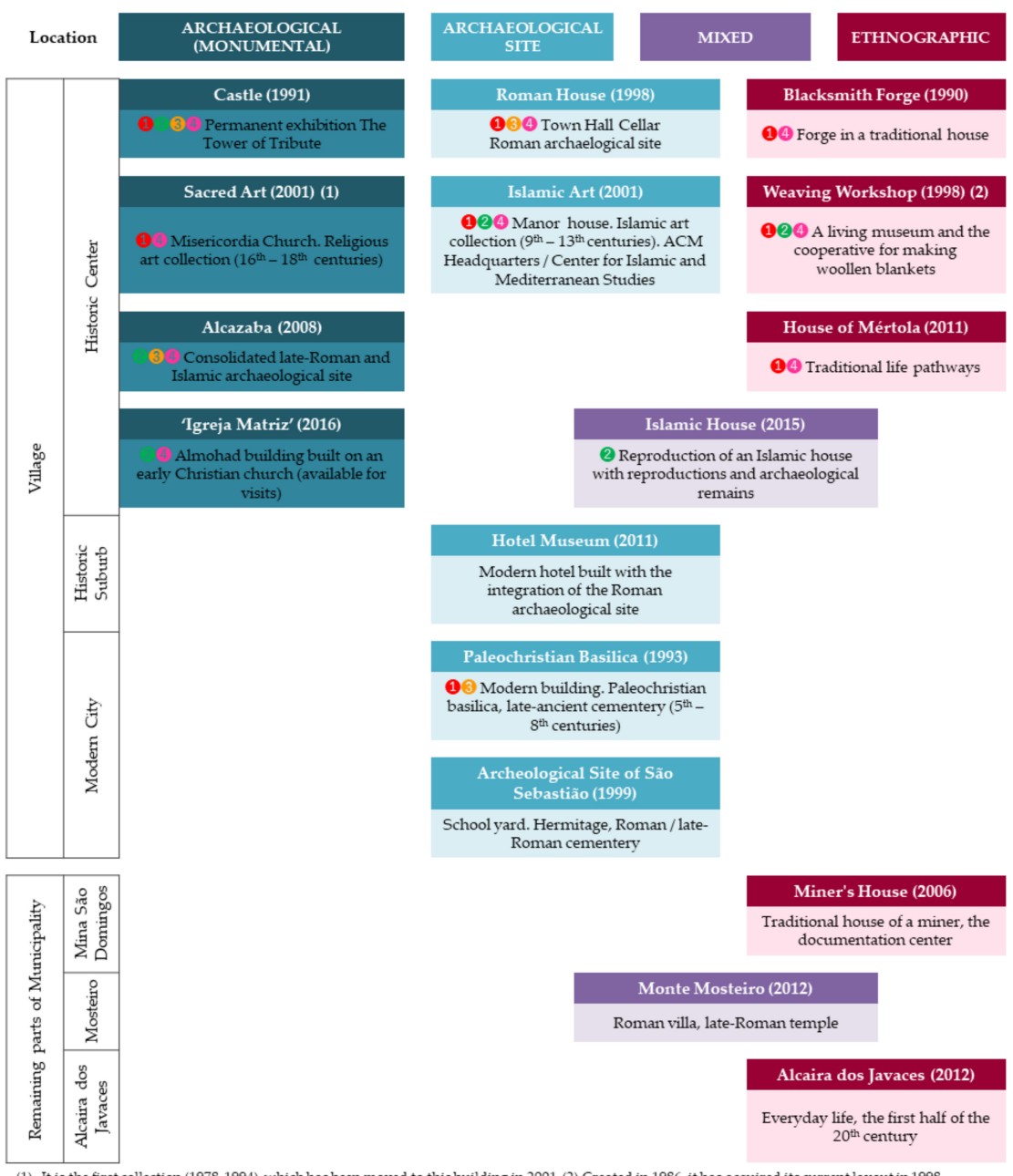

**Figure 4.** Museum centres of the Mértola Museum and urban routes. Source: [72,78–80].

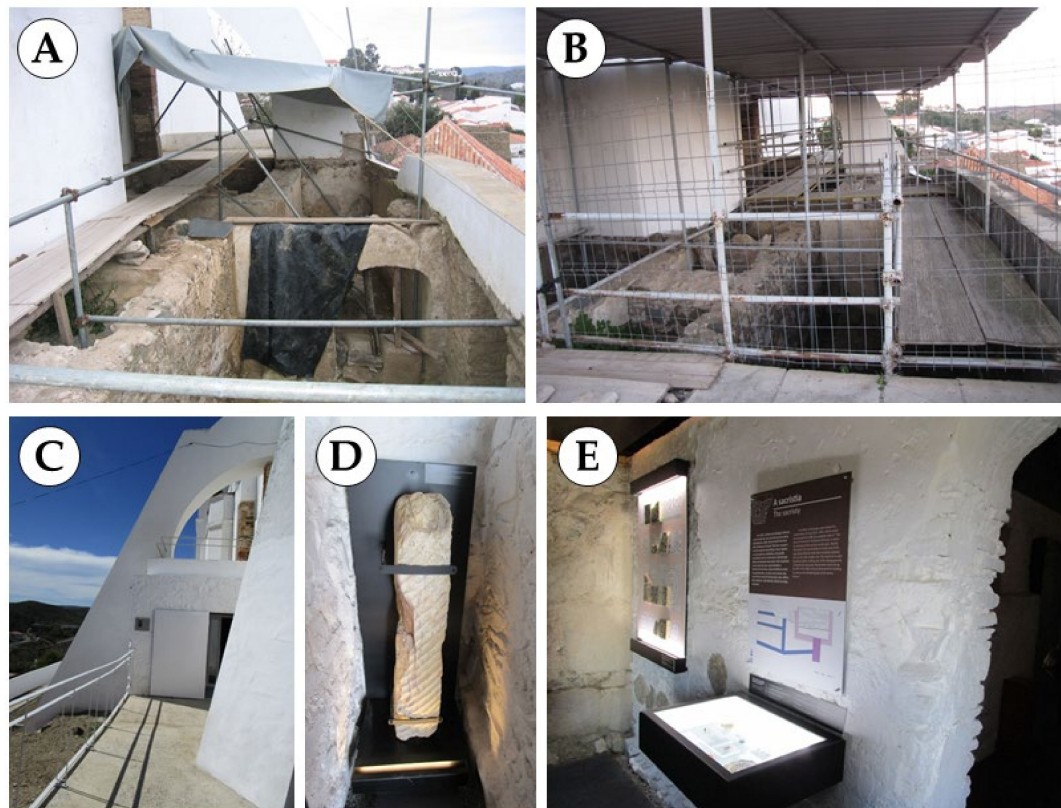

**Figure 5.** Images A and B show archaeological intervention of the early Christian temple in the basement of the Igreja Matriz (E-S side), 28 December 2004 (**A**) and 4 February 2013 (**B**). The nodal point of the museum (**C**), seen from the basement, and interpretation of the remains inside (**D**,**E**), 8 April 2016. The images highlight the enormous work of heritagisation, its slowness, and its cost.

In 2002, after 25 years of PCP ascendency, the Socialist Party (hereinafter PS) came to power in the CMM and a period foregrounding the value to tourism of Mértola's heritage was initiated. This process was based on the conservation and recovery of heritage (involving high costs and low profits).

Although its museological underpinnings were initiated years earlier, the Museum of Mértola was formally established in 2004 by the CMM, in response to abnormalities in the management structure, which prevented its inclusion in the Portuguese Museum Network [73], with scientific specialist responsibility being delegated to the CAM [78]. Efforts to diversify the range of offers from the museum were set in motion from 2006 (Figure 4). In 2001 (that is, before the political shift of power in municipality), the Islamic Festival of Mértola (hereinafter FIM) was inaugurated, organised by the CAM under the auspices of the CMM. A biennial festival taking place over 3–4 days in May. With the accession of the PS, the FIM became an important element in getting the town noticed on the tourist circuit, with the help of media promotions and links to similar events [73].

The tourism-oriented heritage organised within the town included guided visits and themed walks around the centre, leading from node to node of the outdoor museum (Figure 4). The routes were managed by the Tourist Information Centre, which was dependent on the CMM [81], in collaboration with the CAM.

Next, 2004 saw the creation of Merturis, a publicly owned enterprise with the objective of making the most of tourism opportunities with the municipality through the development of products, the projection of an image, and the implantation of strategies to attract, incentivize, and retain tourism-oriented businesses, which would consequently provide local employment [82]. Public company auditing by the Portuguese government led to its dissolution in 2015, without having achieved its objectives.

Following the dissolution of Merturis, the promotion of Mértola passed to the "Visit Mértola" web portal [79], a collaboration between the CMM, the Serrão Martins Foundation (in representation of the São Domingos mines), the Mértola Museum, and Visit Portugal, with the exclusion of the remaining local actors, focussed on advertising the range of tourist activities around the municipality.

The most notable initiative of the CMM has been the nomination of Mértola for inclusion in the "Tentative List of World Heritage Sites in Portugal" (hereinafter LIPMP) drawn up by the National Commission for UNESCO, as a first step towards recognition as a UNESCO World Heritage Site (hereinafter WHS). The candidacy of Mértola was based on three of UNESCO's ten selection criteria, (following UNESCO's own numbering and descriptions) [69], namely:

Criteria ii.　"to exhibit an important interchange of human values" (cultural exchange)—the evidence of diverse civilisations in Mértola, visible in the organisation, architecture, archaeological remains, and traditions of the 'vila' (with special emphasis on the Roman, late antiquity and Islamic periods).

Criteria iii.　"to bear a unique or at least exceptional testimony to a cultural tradition or to a civilization which is living or which has disappeared"—early Christian remains.

Criteria iv.　To be an outstanding example of a type of building, architectural or technological ensemble, or landscape, which illustrates (a) significant stage(s) in human history—the remarkable strategic location of the town in terms of defence and river transport (the castle, city walls, and 'Torre del Río').

The proposal was presented to the National Commission in June 2016 and was initially rejected (Int2). Nevertheless, their recommendations were taken as a positive response (Int1, Int2, Int4), and on 1 July 2017, without these being taken up, Mértola's candidacy was accepted and the town was included in the LIPMP [69].

### 3.2. Evolution of Tourist Activity in the Municipal Context

Taking the number of visits as an indicator of the success of the PMVM, the Mértola Museum has experienced ups and downs (Figure 6). The turning point was the first FIM (2001), which saw the number of visitors increase by 72.10%. The standoff between the CAM and the CMM led to a period of stagnation (2004–2008), with growth returning once relations had been re-established. No increase in visitor numbers can be detected as a result of the town's inclusion on the LIPMP. The Mértola Museum receives more visits than any of the 24 museums in Baixo Alentejo, representing 32.92% of the total within the subregion in 2018, and 62.46% in 2017 (a FIM year).

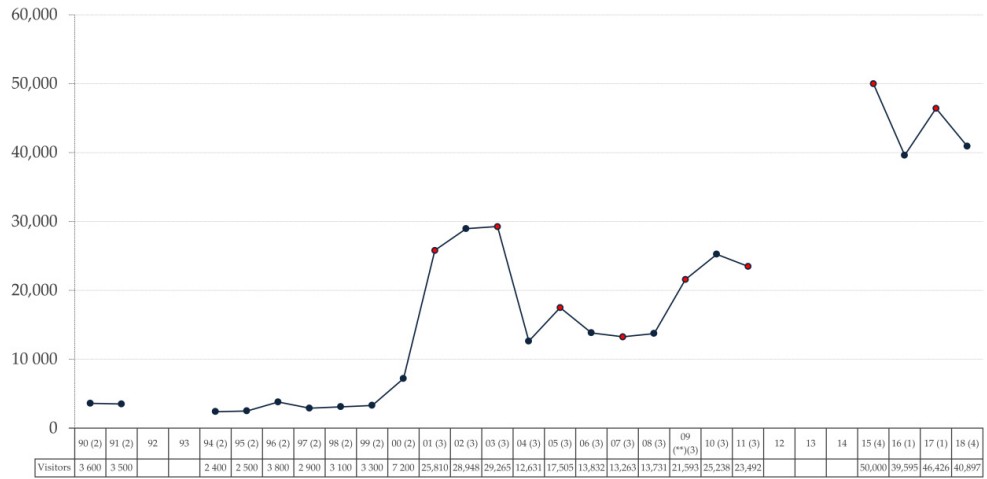

(*) The "registered visitors" do not coincide with the real ones, since it is an open set, but it has 3 registration points:
Igreja Matriz (1), PIT (2), The Tower of Tribute (3). Historical set (4). (**) Until September.
In years indicated in red the FIM is celebrated.

**Figure 6.** Visitors to the Mértola Museum, 1990–2018 (*). Source: CMM visitor data taken from: [67,75,78,83,84].

The income estimated for the Mértola Museum is 1.11 €/visitor in 2012 [83], which represents 3.70% of the CMM's spending in culture and sport, including the museum [67].

The pattern of tourism over time is reflected in the official statistics [18] (p. 1788). The number of nights spent in tourist accommodation in the municipality showed a steady growth (Figure 7) between 2013 and 2018 [67], peaking in the years in which the FIM was held and 2018 (in which the number of guests reached 69.88% of museum visits). Its share of overnight stays within the subregion went from 6.11% (2013) to 14.11% (2018), taking the municipality from fifth place to second. The average length of stay was 1.8 days (2018) [29].

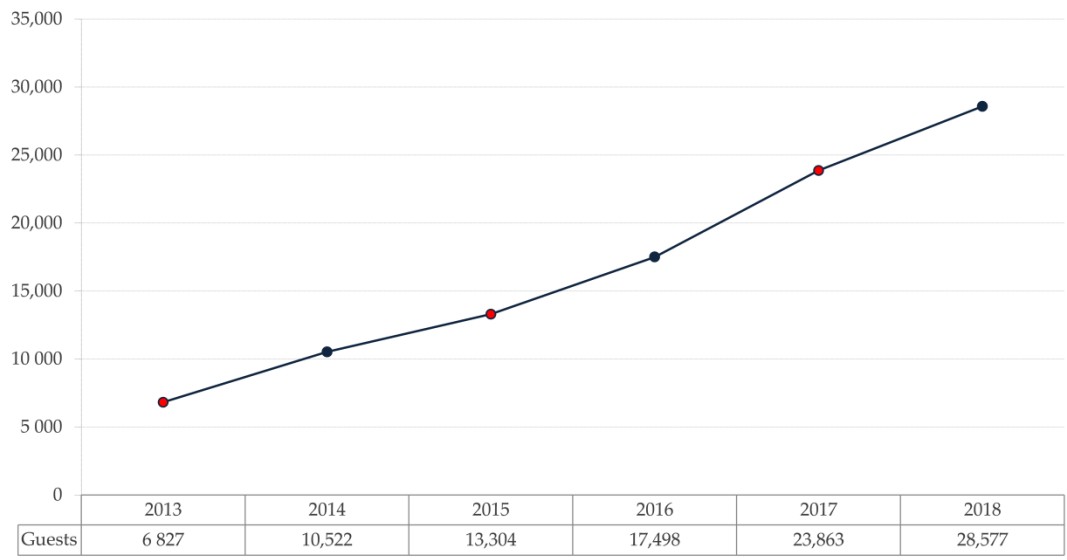

|  | 2013 | 2014 | 2015 | 2016 | 2017 | 2018 |
|---|---|---|---|---|---|---|
| Guests | 6 827 | 10,522 | 13,304 | 17,498 | 23,863 | 28,577 |

In years indicated in red the FIM is celebrated.

**Figure 7.** Guests in accommodation in the Municipality of Mértola, 2013–2018. Source: [67].

The promotion of tourism by the Mértola Municipal Chamber (CMM) began in 2002 [85], reaching 24 places of accommodation in 2020, the first of which was registered in 2008 (Figure 8) [63]. A range of accommodation options have become available [63]: 15 local accommodation points, 7 companies in the rural tourism sector, and 2 hotel establishments, accounting for 32.54% of places.

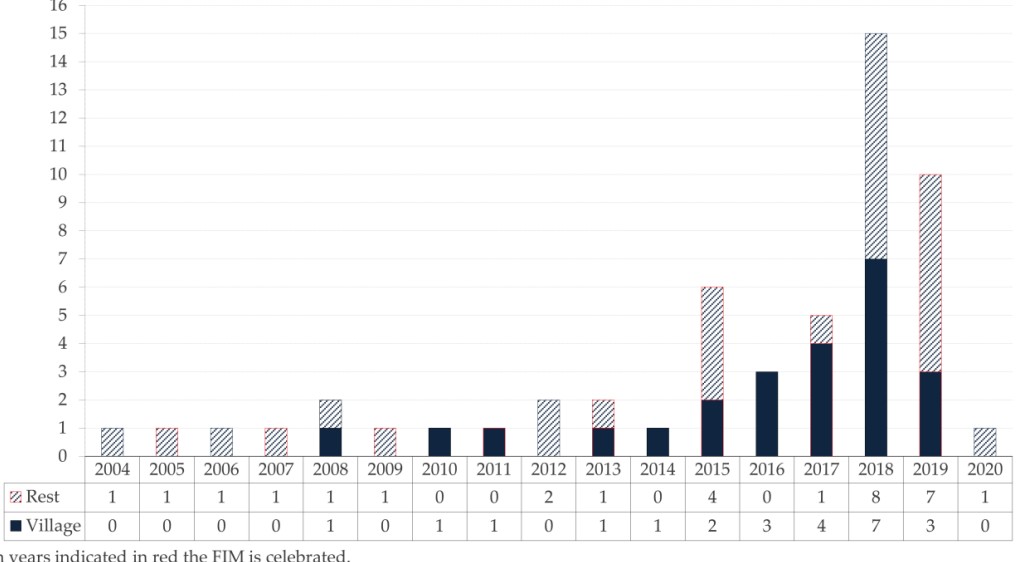

|  | 2004 | 2005 | 2006 | 2007 | 2008 | 2009 | 2010 | 2011 | 2012 | 2013 | 2014 | 2015 | 2016 | 2017 | 2018 | 2019 | 2020 |
|---|---|---|---|---|---|---|---|---|---|---|---|---|---|---|---|---|---|
| ▨ Rest | 1 | 1 | 1 | 1 | 1 | 1 | 0 | 0 | 2 | 1 | 0 | 4 | 0 | 1 | 8 | 7 | 1 |
| ■ Village | 0 | 0 | 0 | 0 | 1 | 0 | 1 | 1 | 0 | 1 | 1 | 2 | 3 | 4 | 7 | 3 | 0 |

In years indicated in red the FIM is celebrated.

**Figure 8.** Evolution of the tourist accommodation register in the village and in the rest of the municipality of Mértola, 2008–March 2020. Source: [63].

In total, at the time of writing, there were 295 beds available in the town (Figure 9). It was predominated by small establishments and only hotels can accommodate groups (≥44 beds).

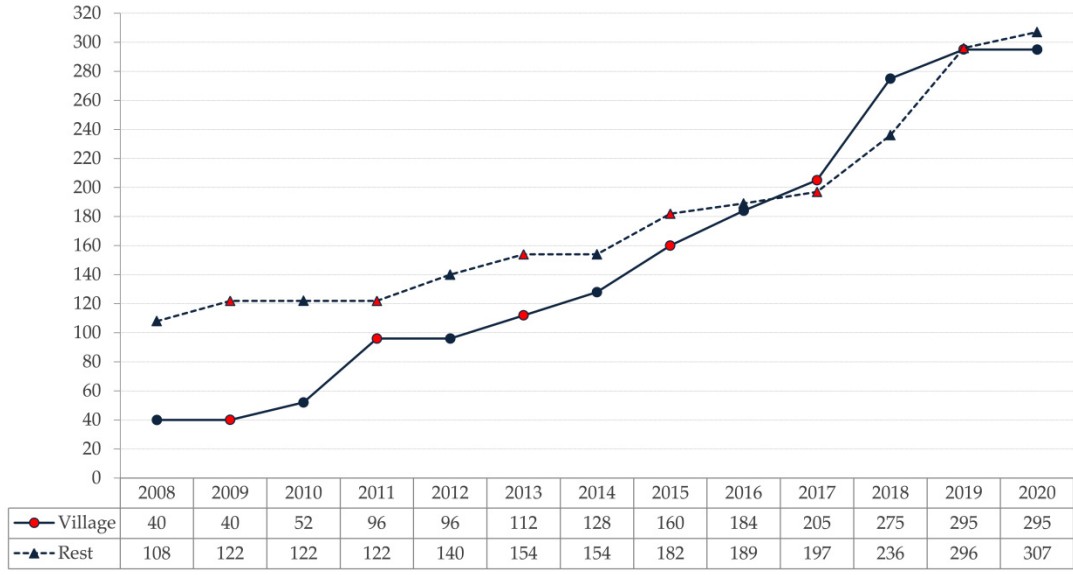

| | 2008 | 2009 | 2010 | 2011 | 2012 | 2013 | 2014 | 2015 | 2016 | 2017 | 2018 | 2019 | 2020 |
|---|---|---|---|---|---|---|---|---|---|---|---|---|---|
| Village | 40 | 40 | 52 | 96 | 96 | 112 | 128 | 160 | 184 | 205 | 275 | 295 | 295 |
| Rest | 108 | 122 | 122 | 122 | 140 | 154 | 154 | 182 | 189 | 197 | 236 | 296 | 307 |

In years indicated in red the FIM is celebrated.

**Figure 9.** Evolution of the number of places in tourist accommodation in the village and in the rest of the Municipality of Mértola, 2008–March 2020. Source: [63].

Merturis was first to offer activities in 2004. At the time of writing, in 2020, there were seven companies based in the town, including six focussing on tourist activities and one travel agency, five of which offer cultural activities—three solely cultural and two in combination with other types.

There were also 20 restaurants of varying types in the town (55.56 of the total in the municipality) with 1138 seats (60.34% of the municipal total) [79]. A total of 12 establishments had a seating capacity ≥50 amounting to 1083 seats.

An increase in the number of shops selling artisanal products and/or souvenirs (four) can be noted at points of access to the historical old town and in workshops within its walls (two), manifestations of heritagisation, in particular training courses for recovering of traditional crafts (Int4).

The companies involved in heritage and tourism can be divided into three types (Int4, Int5):

- Entrepreneurship: small start-ups with no background in the field (specialist public employees) or self-employment deriving from training either professional or at a university.
- Sectorial diversification/income supplement: small-scale initiatives aimed at the diversification of typical products or noncorporate employment, mostly at local accommodation.
- Investment: internal investment concentrated on hospitality and accommodation by agents in other productive sectors setting up separate businesses and external companies mainly focused on investment funds and real estate. These are companies with complex business structures.

There is a predominance of personal investment (Int1, Int4, Int5), and the co-financing of initiatives with European grants managed by local action groups is scarce, and generally limited to institutionally managed investment such as the CMM, CAM, and ADPM (Int4). Some specific projects have been financed, with seven initiatives in town receiving support between 1996 and 2015, namely four connected to tourist accommodation, two restaurants, and one tourism activities business. The tendency is to finance investment projects beyond the reach of local entrepreneurs.

Tourism is diversifying the Mértola economy (Figure 10). The two sectors with the highest number of companies are the primary and service sectors. The "accommodation, restaurant, and similar businesses" sector represented 12.47% in 2017, demonstrating a higher degree of stability than other activities.

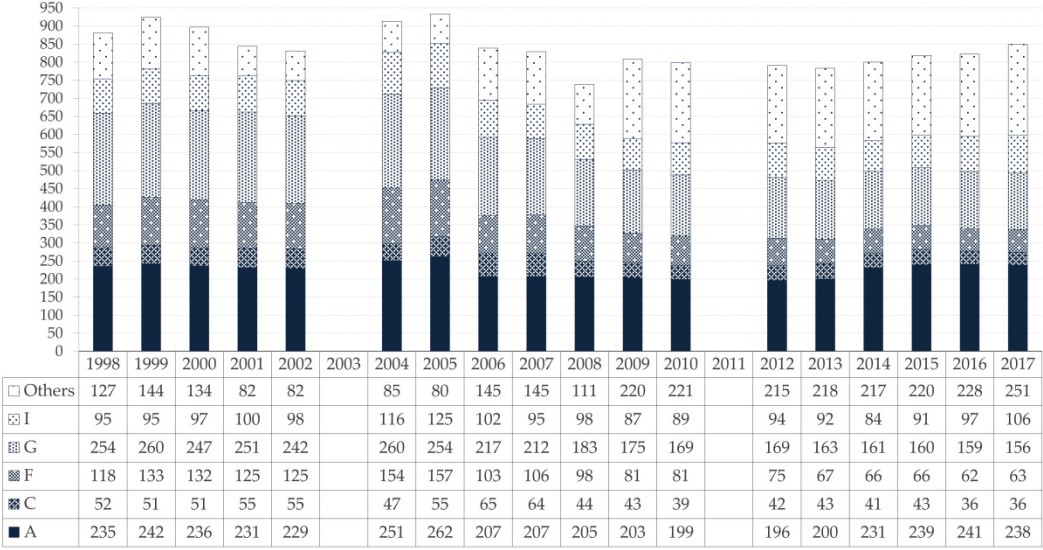

| | 1998 | 1999 | 2000 | 2001 | 2002 | 2003 | 2004 | 2005 | 2006 | 2007 | 2008 | 2009 | 2010 | 2011 | 2012 | 2013 | 2014 | 2015 | 2016 | 2017 |
|---|---|---|---|---|---|---|---|---|---|---|---|---|---|---|---|---|---|---|---|---|
| ☐ Others | 127 | 144 | 134 | 82 | 82 | | 85 | 80 | 145 | 145 | 111 | 220 | 221 | | 215 | 218 | 217 | 220 | 228 | 251 |
| ⊠ I | 95 | 95 | 97 | 100 | 98 | | 116 | 125 | 102 | 95 | 98 | 87 | 89 | | 94 | 92 | 84 | 91 | 97 | 106 |
| ⊞ G | 254 | 260 | 247 | 251 | 242 | | 260 | 254 | 217 | 212 | 183 | 175 | 169 | | 169 | 163 | 161 | 160 | 159 | 156 |
| ▦ F | 118 | 133 | 132 | 125 | 125 | | 154 | 157 | 103 | 106 | 98 | 81 | 81 | | 75 | 67 | 66 | 66 | 62 | 63 |
| ▦ C | 52 | 51 | 51 | 55 | 55 | | 47 | 55 | 65 | 64 | 44 | 43 | 39 | | 42 | 43 | 41 | 43 | 36 | 36 |
| ▪ A | 235 | 242 | 236 | 231 | 229 | | 251 | 262 | 207 | 207 | 205 | 203 | 199 | | 196 | 200 | 231 | 239 | 241 | 238 |

A: Agriculture, animal production, hunting, forestry and fishing; C: Processing industries; F: Construction;
G: Wholesale and retail trade, vehicle repair; I: Accommodation, catering or similar.

**Figure 10.** Evolution of the number of companies by type of activity in the Municipality of Mértola, 1998–2017. Source: [67].

In terms of business volume (Figure 11), it is notable that the service sector has increased while the primary sector has stagnated. After a period of slowdown brought on by the international economic crisis, the hospitality sector, especially accommodation and restaurants, experienced significant growth from 2014 with an accumulated increase of 81.34% between 2014 and 2017. Total income per tourist bedroom in 2017, as an FIM year, rose to 1474 €, which was an increase by 80.19% over 2013, while the average spending of guest/day was 10.26€ [67] due to the abundance available of local accommodation.

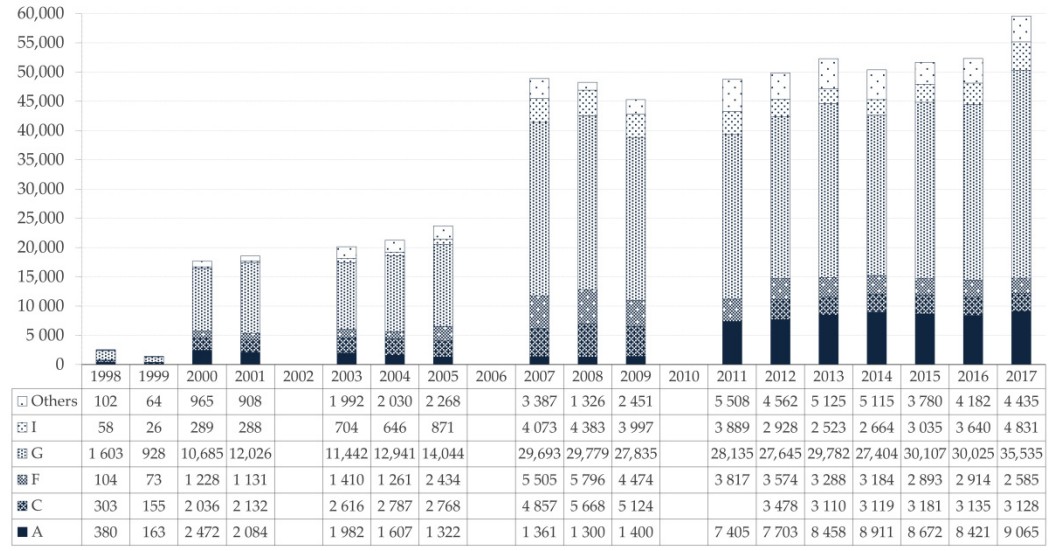

| | 1998 | 1999 | 2000 | 2001 | 2002 | 2003 | 2004 | 2005 | 2006 | 2007 | 2008 | 2009 | 2010 | 2011 | 2012 | 2013 | 2014 | 2015 | 2016 | 2017 |
|---|---|---|---|---|---|---|---|---|---|---|---|---|---|---|---|---|---|---|---|---|
| ☐ Others | 102 | 64 | 965 | 908 | | 1 992 | 2 030 | 2 268 | | 3 387 | 1 326 | 2 451 | | 5 508 | 4 562 | 5 125 | 5 115 | 3 780 | 4 182 | 4 435 |
| ⊠ I | 58 | 26 | 289 | 288 | | 704 | 646 | 871 | | 4 073 | 4 383 | 3 997 | | 3 889 | 2 928 | 2 523 | 2 664 | 3 035 | 3 640 | 4 831 |
| ⊞ G | 1 603 | 928 | 10,685 | 12,026 | | 11,442 | 12,941 | 14,044 | | 29,693 | 29,779 | 27,835 | | 28,135 | 27,645 | 29,782 | 27,404 | 30,107 | 30,025 | 35,535 |
| ▦ F | 104 | 73 | 1 228 | 1 131 | | 1 410 | 1 261 | 2 434 | | 5 505 | 5 796 | 4 474 | | 3 817 | 3 574 | 3 288 | 3 184 | 2 893 | 2 914 | 2 585 |
| ▦ C | 303 | 155 | 2 036 | 2 132 | | 2 616 | 2 787 | 2 768 | | 4 857 | 5 668 | 5 124 | | | 3 478 | 3 110 | 3 119 | 3 181 | 3 135 | 3 128 |
| ▪ A | 380 | 163 | 2 472 | 2 084 | | 1 982 | 1 607 | 1 322 | | 1 361 | 1 300 | 1 400 | | 7 405 | 7 703 | 8 458 | 8 911 | 8 672 | 8 421 | 9 065 |

A: Agriculture, animal production, hunting, forestry and fishing; C: Processing industries; F: Construction;
G: Wholesale and retail trade, vehicle repair; I: Accommodation, catering or similar.

**Figure 11.** Evolution of business volume (thousands of Euros) of companies by type of activity in the Municipality of Mértola, 1998–2017. Source: [67].

In 2017, the hospitality sector employed 11.88% of the total workforce. Since 2014, the trend has been upwards, with an increase of 26.54%, although it is still not the major sector in terms of employment (Figure 12).

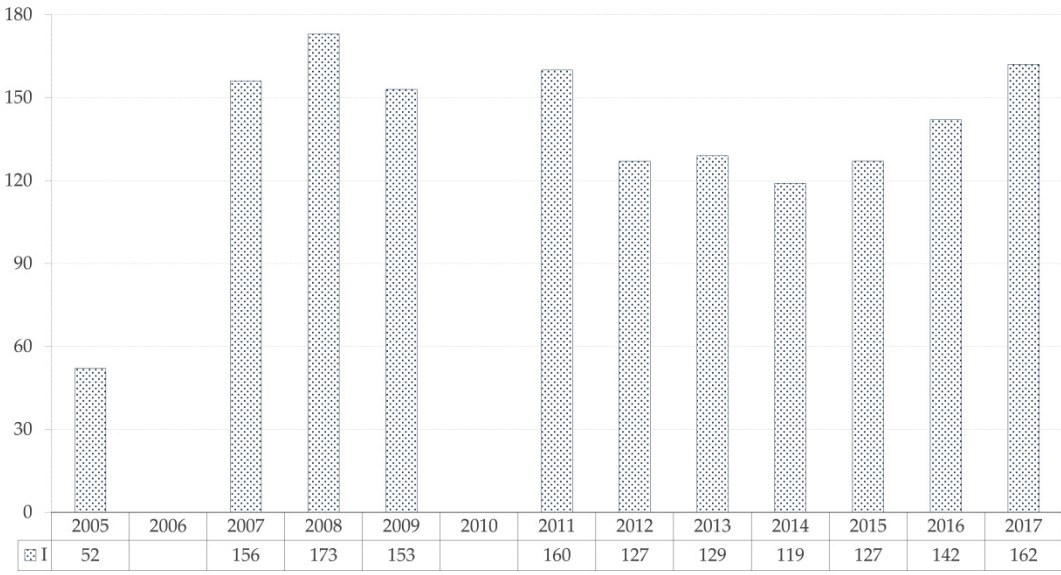

| ⊠ I | 2005 | 2006 | 2007 | 2008 | 2009 | 2010 | 2011 | 2012 | 2013 | 2014 | 2015 | 2016 | 2017 |
|---|---|---|---|---|---|---|---|---|---|---|---|---|---|
| | 52 | | 156 | 173 | 153 | | 160 | 127 | 129 | 119 | 127 | 142 | 162 |

I: Accommodation, catering or similar.

**Figure 12.** Evolution of employment in accommodation and catering companies in the Municipality of Mértola, 2005–2017. Source: [63].

The largest single employer in the municipality is the CMM, which has 316 employees [86], or about 10% of the local workforce. Eleven of these employees work in the area of culture, i.e., representing 3.48% of the CMM total, and 14 in the area of tourist information and museums, namely 4.43% of the total [86]. It is not possible to give the corresponding number of workers for the CAM as this is the variable in terms of the projects and the incorporation of workers and researchers is seasonal. The skilled workforce, university graduates and technical specialists, is employed mainly in the CMM and CAM. The qualifications are, in part, the result of the training programmes (EPJBC, ALSUD) [77].

According to the statistics, the rural exodus continues, with a decrease of 37.26% in the population between 1991 and 2019 (Figure 13). However, the decrease slackened off between 2010 and 2019, with the period 2018–2019 showing the least loss across the yearly intervals at the rate of −0.81%. The net balance is negative, although the rate has reduced from 2015 going from −1.15% to −0.38 in 2018.

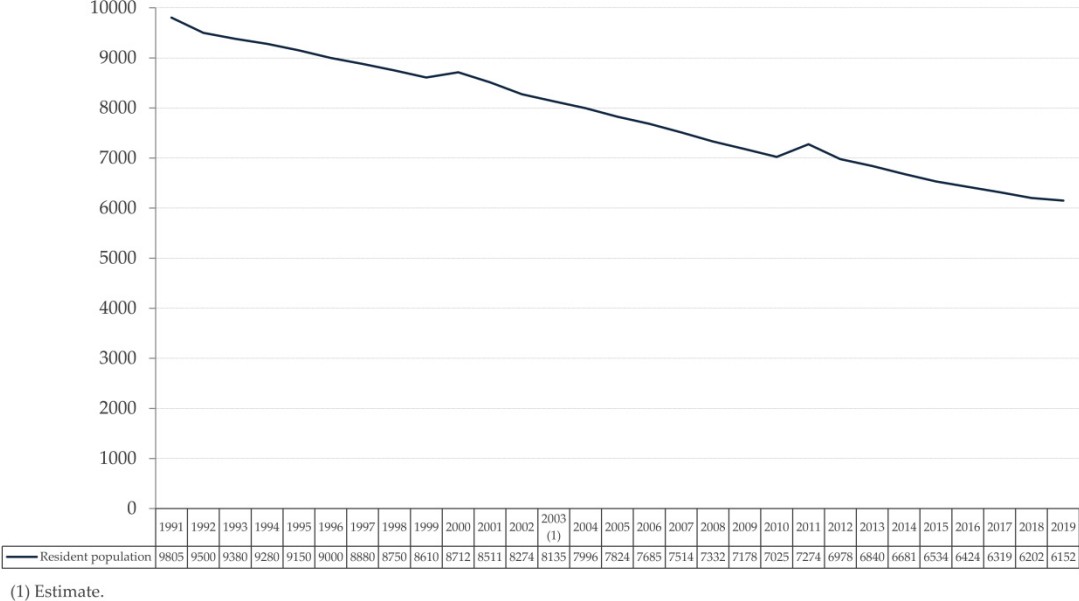

| | 1991 | 1992 | 1993 | 1994 | 1995 | 1996 | 1997 | 1998 | 1999 | 2000 | 2001 | 2002 | 2003 (1) | 2004 | 2005 | 2006 | 2007 | 2008 | 2009 | 2010 | 2011 | 2012 | 2013 | 2014 | 2015 | 2016 | 2017 | 2018 | 2019 |
|---|---|---|---|---|---|---|---|---|---|---|---|---|---|---|---|---|---|---|---|---|---|---|---|---|---|---|---|---|---|
| Resident population | 9805 | 9500 | 9380 | 9280 | 9150 | 9000 | 8880 | 8750 | 8610 | 8712 | 8511 | 8274 | 8135 | 7996 | 7824 | 7685 | 7514 | 7332 | 7178 | 7025 | 7274 | 6978 | 6840 | 6681 | 6534 | 6424 | 6319 | 6202 | 6152 |

(1) Estimate.

**Figure 13.** Resident population in the Municipality of Mértola, 1991–2019. Source: [67].

The process of heritagisation has taken place at the same time that residents moved out of the historical town centre to take up residence in the new part of town or to leave altogether. This outflow contributed to the deterioration of the centre, which is taking considerable time, money, and effort to restore. In order to prevent the emptying of the historical centre and to encourage people to return, the CMM established a package of measures to support the rehabilitation of local heritage [78], aimed at: (a) restoring buildings for use by the municipal services, such as the CAM, the museum, and so on; (b) the promotion of events and economic activities, specially the FIM and similar celebrations; and (c) social housing.

### 3.3. Assessment of the Actions in the Context

With the restoration of democracy to Portugal in 1974, the notion of heritagisation was popularised and the concept of so-called historical value began to take place [73]. The previous patrimonial declarations responded to the protection of the object in 1910 and to the national exaltation, namely the dictatorship.

The process of heritagisation in Mértola would be carried out in a social, political, and ideological context [87]. The notion of "integrated development founded on heritage resources" [75] (p. 32) was taken up, in which heritage was understood as "collective memory", and the overall objective was local development through the involvement of the community [76]. At the same time, social and cultural capital were recognised as the foundations on which the project was built [10,11].

Since 1980, the interaction between public and private institutions—the CMM on the one hand and the CAM and ADPM on the other—has been a complex process. The sheer scale of the conservation involved became a major challenge [24] (Figures 5 and 14), particularly in view of the lack of ongoing funding [43]. The CMM jointly funded activities, the museum, and provision of space, while the CMM, CAM, and ADPM sought external funding at regional, state, or community level [88] for intervention/research projects [89].

The foundation for the project was an intricate museographic project, based on the notion of a "community museum" [29] (p. 1), envisioned as an educational tool for exploring identity and heritage at the service of humankind now and into the future [90]. The PMVM received recognition [74] for its good practice, and attracted considerable national and international attention as a result of its scientific content, which went beyond the university system, methodology, endogenous orientation, the inclusion of local residents and their concerns, and the divulgation of the results [77,78]. It also took on the task of training locals [76], beginning with courses for specialist personnel sponsored by the ADPM (1978–1985) [75]. The PMVM incorporated training for the local population [87], which began with ADPM courses for technical staff (1978–1985) and later became the Bento de Jesús Caraça Professional School (hereinafter EPJBC) [78].

In a peripheral territory incorporating different elements—such as a natural park, cultural heritage, and outstanding landscapes—it was deemed necessary to create, promote, and sell products [29], overcoming the limitations imposed by its location on the periphery so as to make these viable [6]. As a consequence, since 2002, the CMM's cultural and tourism policy focussed on [83]:

(a) Stimulation of tourism packages around different approaches, i.e., heritage, nature, active lifestyles/sport, gastronomy, industrial/mining heritage, and hunting.
(b) Expansion of initiatives throughout the municipality, not only the town.
(c) Creation of quality-focussed products and image for commercialization.
(d) Increased involvement of local population.
(e) Active search for private investment.

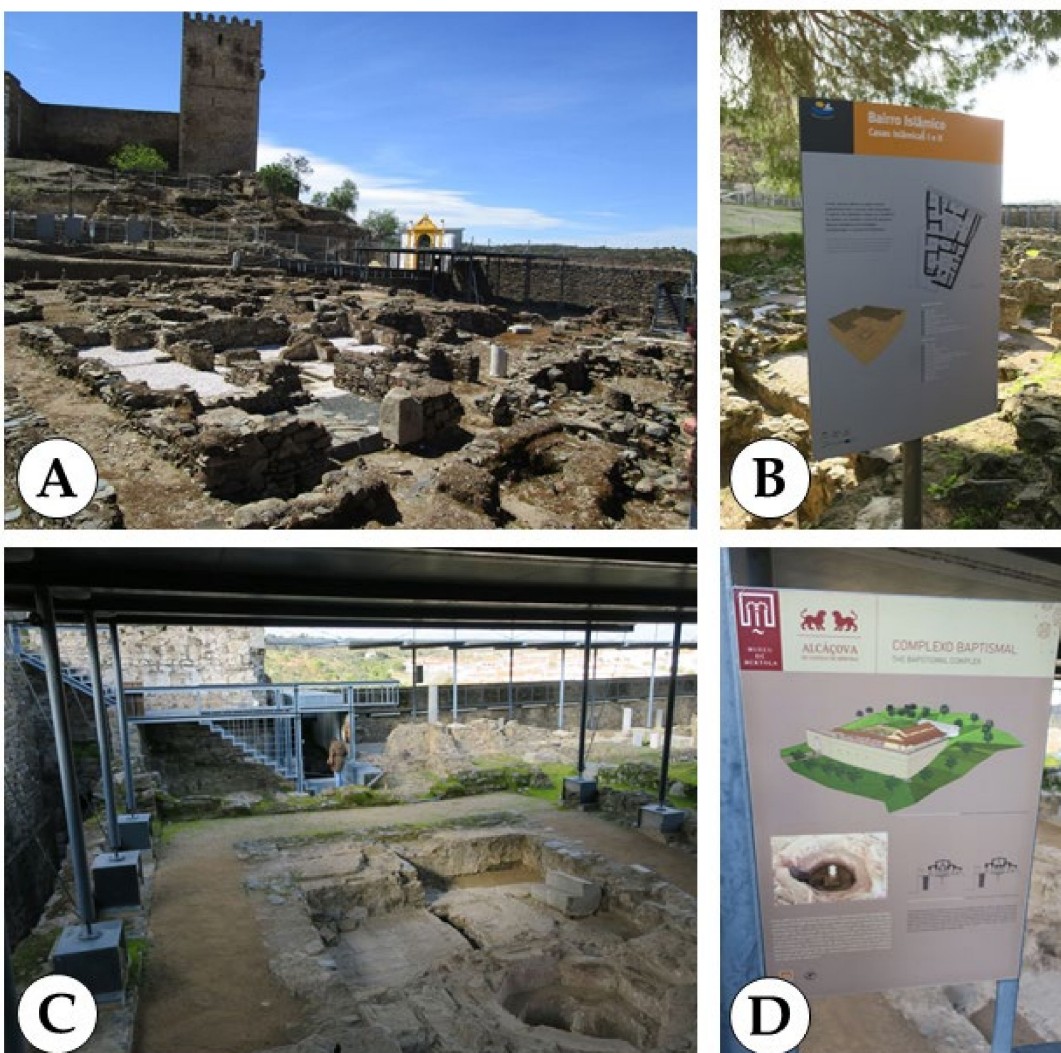

**Figure 14.** Alcazaba of Mértola (the nodal point of the museum). The Excavation of the Islamic Quarter with the castle and the Tower of Tribute in the background (**A**) and the interpretation signage (**B**), 9 February 2019. The intervention to consolidate the baptismal complex (**C**) and the interpretation signage (**D**), 8 April 2016. The images show the great space of intervention and the process of conservation of the archaeological heritage.

Although the museum was doubtlessly the maximum expression of heritagisation, and the vila as its most important resource, from a visitor's perspective, it was recognised as of limited interest [91], with preservation and research taking precedence over tourism [29] (Figure 14). It did, however, embody a variety of diverse perspectives, particularly socio-cultural, religious, military, and other activities, such as a research centre and traditional crafts, which, considered jointly, enhanced its potential as the tourism attraction [41]. The establishment of these routes around town helped to reinforce the idea of a unified collection of elements [92], although their physical dispersion made it difficult to integrate all of them into a whole.

The FIM has proposed as an innovative event, in line with authenticity, to promote Islamic heritage and local history [93,94], through a series of cultural and artistic activities and scientific conferences, all of which were held within the town walls to revitalize the historic centre and involve residents. Community-based events such as this are important to the life of peripheral areas [29,95], and can become important cultural attractions [41], capable of attracting more than 20,000 visitors.

Merturis has not achieved its objectives, and Visit Mértola, dedicated to all tourist activities in the municipality, has not managed to increase the necessary participation of private capital in the promotion [96] or public–private collaboration [58], nor has it created a unique image of the destination or the brand [97,98].

Finally, in the preparation for its candidacy that followed, the town pursued its bid to become a WHS in the hope that such a declaration would kick-start tourism in the area [99]. The view of the CAM was the approach that represented a political project related to the promotion, rather than technical matters (Int2, Int4), and pointed out the lack of a strategy, whilst acknowledging its potential.

### 3.4. Positions Indicated Among Stakeholders

Each of the actors has their own and "constructed reality" [100] (p. 79) interests [46], given voice by the prevailing discourse and shaped by a representative framework [101]: the CAM takes a conservationist position, while the CMM follows a more commercial view. In short, the conflicting interests represent the classic trade-off between heritage, conservation or curatorship, and tourism [43]. Each discourse aligns itself with particular ideological projects and modes of understanding the progress, especially heritagisation in terms of the safeguarding the cultural identity of the local inhabitants and the promotion of tourism that bring tangible economic benefits to the local population.

For the CAM, their conservationist position is motivated by ideological beliefs clustered around collectivism and egalitarianism [73]. Both public (CMM) and private (CAM/ADPM) initiatives should be aligned on issues of cultural identity [102], for which heritagisation is the means to contribute to "community development". In this view, cultural values should always prevail and the tenets of tourism are disparaged as commercialism, where profit is prized above the inherent value of heritage [103], and there is a reluctance to fix prices [42,104]. Although this discourse has developed over time [75], tourism remains a result rather than an end in itself, something that could contribute to maintaining the local inhabitants [73] and their identity [89,102]. The interests of the CAM are not in tourism but in heritagisation. However, as this needs to be financed (Int2), they opt for small-scale initiatives "so as to avoid multinational hotel chains" [89] (p. 1).

By contrast, the commercialist discourse of the CMM is linked to a view of local development as a coming together of endogenous and exogenous, public and private forces—endogenous foundations with an outwards projection focusing on searching for investors and finance. The emphasis is on tourism as generator of wealth, with the role of the CMM being that of curating and promoting, while private enterprises take responsibility for tourism initiatives. In this vision, tourism becomes a development strategy for stimulating cash flow and bringing in sufficient returns to finance heritage conservation, but always on the principle that the "user pays" [42]. Nevertheless, the tourism-focused view of development "runs the risk of neglecting other important factors and processes" [26] (p. 740). Although politicians insist that tourism could reduce regional disparities, expectations tend to be over-optimistic [18,30].

The CMM's local policies regarding the process [82] have generated informal agreements between public and private actors, with hegemonic discourses that "can constitute a 'regime' that in turn shapes local policy-making" [58] (p. 25). A balance needs to be reached between the policies of heritagisation and tourism without losing sight of the issue of sustainable growth [105] and creative construction/destruction [106].

Stakeholders can play a significant role. They can become empowered and improve well-being over the long-term [4], acquiring agency as a result of their own influence and through the relationships developed among themselves [47,99]. Collaboration between stakeholders has thus become a major issue [49].

The main stakeholders in the heritagisation process are the CMM and the CAM, also previously known as the ADPM. They act at the same level, each has its area of expertise, but they share the same discourse, ideological programme, and interests, in which the principles of cooperation [46] and collaboration [27,107] are paramount. An early issue was that of leadership [108], a role initially filled

by the first democratically elected mayor (Int3, Int4), who carried out the role of managing relationships between the interested parties [4]. After the death of the mayor in 1982, the ADPM, replaced by the CAM in 1988 and thereafter, took on the scientific and intellectual leadership, their authority being recognised by the CMM. The cooperation successfully initiated a large-scale process of heritagisation (PMVM).

When the socialists were voted in to govern the CMM in 2002, the conflicting perceptions and issues of discourse [57,101] between the political parties and their leaders (Int2) produced a rift. This led to a change in relations between the agencies, and hierarchies began to appear through the CMM, taking over the leadership alongside rivalries and disagreements [57]. In 2004, the CMM put into action a plan to amplify the number of stakeholders to include Merturis and the Serrão Martins Foundation (a foundation set up for the conservation and projection of the São Domingos Mine visitor centre, which is a short drive from Mértola.), limiting the power of the CAM/ADPM and acting as a counterweight. At the same time, the CMM enlarged its own power by taking over the running of the Mértola Museum, albeit deferring to the authority of the CAM in scientific matters (Int4). These tensions manifested themselves in the interruption of the heritagisation process, the halt to the training programme, and the poor outcomes in terms of tourism.

When, in 2008, the socialist mayor left the post, contact between the CMM and CAM/ADPM was resumed, and although there remained a gap between their viewpoints, "there was a new injection of life in the heritage question" (Int4). The rapprochement between the two sides reinvigorated performance and the achievement of objectives [43,47], reactivating the processes of heritagisation and promotion of tourism, and setting in motion again the training programme under the stewardship of a new entity, named as the ALSUD Training School [109]. The renewed impetus to attract tourism brought a new stakeholder in to the frame, the Association of Business Owners (Int5).

Criticism of the leading figures within the CMM by the CAM includes "not being up to the task (...) conservation is not a course of action" (Int2) and "wanting to live only off tourists" (Int4). The CAM also underlines that "the CMM shouldn't be doing everything (...) and the private sector [referring to the CAM/ADPM] should also be a part of things" (Int2). For its part, the CMM maintains that "the primary objective is the scientific [heritagisation], on the basis of which tourism can be developed, and then in its turn local development" (Int1). The discourse does not attempt to delegitimize the CAM; it recognises its expertise and good practice [83] and its authority in scientific matters (Int1), but it claims for the CMM a role in the management and promotion of tourism, and, given the similarities with electoral campaigning, the projection of the town to the wider world (Int1).

The actors were aware of the dangers inherent in a lack of coordination and collaboration, and recognised partisanship as the main obstacle to achieving this [83]. It was clear that strategies for improving relations were needed [46], along with a network for facilitating decision-making in matters concerning the development of tourism [110,111], but neither side, it seemed, was willing to take the first step towards opening up the dialogue [43]. The business sector sensed a political/ideological impasse which "meant that [tourism] didn't work" (Int5). According to Da Rosa [109], actors themselves should not be foregrounded but rather the result of their collaboration and the instance the recommendation fell on deaf ears [51]. In some cases, such as that of the FIM, the existence of common interests strengthened relationships, but in others, such as that of the LIPMP project, it amplified rivalries [58].

One thing that appeared in the objectives and discourse of more than one of the actors (Int1, Int3) was the importance of the participation of local residents, given that this was considered crucial to the whole process of development, and a means of avoiding conflict and bringing stability to the projects [112,113]. It lay at the heart of the question of identity and was considered to be closely connected to education and awareness (Int1, Int2, Int4), and to reinforcing the community's confidence to manage its heritage [43,112]. At the start of the heritagisation process, the political affinities of those involved led to the involvement of young people in the project [75], and their participation in the ADPM/CAM (Int4), which can be viewed as kinds of "community heritage groups" [56] (p. 459). In fact, Duarte [73] underlined local empowerment in two respects: (a) the diversification of cultural facilities

and (b) the implementation of mechanisms for the promotion and participation of different social agents. Despite this, starting from 2002, a gradual disaffection of the locals with the archaeological activity, the heritage, and museums began to be noted [73,109]. The CAM put this shift down to "a departure from the original idea on the part of the PS" (Int4), while the CMM blamed it on the fact that the results of the process were not sufficiently visible [83]. A deeper look at the causes is required in terms of discourse, unfulfilled expectations, and stakeholder attitude, among others.

The heritagisation process included certain objectives stated in the PMVM, but the CMM did not develop any specific objectives for strengthening tourism beyond "local development". There was no plan outlining the strategies to be followed, as testified by the absence of an official heritage declaration for the complex and the existence of a Plano de Salvaguarda e Valoriçâo do Centro Histórico da Vila de Mértola [114], a town planning document, revised in 2017, in which the focus was solely on housing-related matters. In order to create a model of governance that enhances tourism sustainability while mitigating negative effects [15], developing viable and temporally and environmentally sustainable attractions [6] it is necessary to define objectives, formulate strategies [57], and implement measures and actions through a participative process. Such a model would also enable the search for finance to palliate the negative effects of peripherality [6], at the same time that innovations in the tourism sector generated new interactions and improved relations between stakeholders, implementing institutional changes [11]. The inclusion on the LIPMP could contribute to this, although it would require a thorough further study.

### 3.5. The Successes, Failures, Results, and Overall Impact of the Processes

The increase in the number of the visitors to the Mértola Museum (Figure 6) indicates its importance and consolidation as a heritage destination [29]. While the confrontation between CAM and CMM has caused stagnation (2004–2008), later, as the collaboration resumed, growth has been observed once again [27,107]. Although there was no increase in the number of the museum visits due to its inclusion in the LIPMP, there was an increase in the number of guests stays (Figure 7).

Tourist activity indicates a certain marked seasonality, as detected in other peripheral spaces [29] with overnight stays concentrated in summer (38.3% in 2018) and May in FIM years, complicating business viability [12]. Nonetheless, there are more incidental trips with a purpose [29], as shown by the increase of overnight stays in the recent years (Figure 7). However, the organisation of individual travel and the predominance of the use of one's own car [33,115] determined connectivity and distance to be key factors [29] with a predominance of national tourism and on the borders, especially on the Spanish side.

When the PMVM was initiated, there has been no tourist offer in the municipality of Mértola. The opening of tourist accommodation in the town has opened since 2008, coinciding with the reestablishment of the contacts between CMM and CAM. Most of the openings coincided with FIM years. The year 2018 provided the turning point due to the expectations generated by the inclusion of the village in the LIPMP. The presence of different types of accommodation demonstrates an adaptation to different markets [24], however, accommodation without internationally recognised quality standards predominates [27].

In order to attract and retain visitors, and so generate income, it is important to be able to offer a range of activities [116]. This offer has appeared since 2004 as a result of the Merturis activity, while enhancing the competition for private businesses (Int5). Once again the effect of the years in which the FIM took place can be seen in the increased demand, alongside the impact of inclusion on the LIPMP, specifically, the founding of two companies in 2017.

Gastronomy is a vital factor in rural and cultural destinations [117]. The offer of Mértola is boosted through traffic breaking up the journey at roadside establishments [29], which again introduces the problem of seasonality.

The institutional context is viewed favourably as a key to development by the companies (Int5), which highlight the promotional efforts of the CMM and the simplification of administrative and

legislative procedures. Entrepreneurship, which is dominant and more dynamic compared to the other types of business, has been driving the range of activities available based on leadership, and opening up new opportunities in the relatively underdeveloped rural tourism sector [118], where there is a little business culture [12]. Nevertheless, at the time of writing, some 51.72% of business volume connected with accommodation and activities was concentrated in the hands of four groups, namely two entrepreneurs with a variety of ventures, a hotel company, and a foreign-owned real estate business.

The situation of deprived areas on the periphery make a flow of investment necessary [119] in a sector of high costs and low returns [12], and where the local public initiative focuses on revitalization/promotion (Int1). Financial and specialist technical support is essential in the long term [12,27] to avoid/limit the ingress of capital from outside.

According to the head offices of the companies involved, outside investment in local real estate has been growing over the last few years, which has had a negative effect on capital accumulation [18]. In addition, the growth in online platforms for managing accommodation is displacing local involvement, to the detriment of the available options [120], and depleting the value added. On the plus side, the platforms make the process of booking rural locations far easier, but the dominance of external operators remains a challenge [12].

The heritage and tourism represent direct and indirect employment opportunities, as well as self-employment [22], one of the chief objectives of rural tourism [118], but the structure of tourism may not be so attractive to the local population [12] due to the limited number of jobs it creates. Jobs are being created within the sector, at the same time that the primary sector workforce is diminishing [29], with an average of 1.53 workers per company, although this varies according to the type of establishment and its capacity.

Employment in this sector is especially susceptible to the effects of seasonality [95], which can be particularly felt in the smaller companies [121]. Problems connected to the lack of a business culture and a prevailing agricultural mentality can also be noted (Int4, Int5), and there are human resources recruitment problems (Int5). Another problem that has been noted is the uprooting of the workforce [15], as many companies are controlled by external groups, although they do create employment. The available data do not indicate whether it is among the most disadvantaged groups that employment is created [122], whether there is any hidden employment, and what the repercussions are for the labour market and unemployment. Exploring these issues would increase our understanding of the processes of local development and power relationships [123].

Rural tourism is conceived of a means of attracting and anchoring a stable population [6,9,18,29]. The slowdown of decline may be related to the dynamism of tourism and the training efforts provided by EPJBC and ALSUD has partly provided the possibility to fill in the lack of essential necessary skills [12]. Nonetheless, it is necessary to increase qualified employment in tourism business to avoid depopulation processes.

While tourism has allowed the historic centre to gain interest with the rehabilitation of traditional buildings (Int2, Int4) as tourist accommodation (13) and second homes [85,124,125], it has resulted in its depopulation and the touristification processes [85]. As a result, the gentrification process has been summed up with the ghettification (Int5) due to the municipal policy generating a complex balance circumstances. There is a risk of theming, such as can be observed in other comprehensive heritage projects, such as that of Óbidos [73].

Among the positive cultural impacts that are worth noting are the reappraisal of the town's heritage, which had been lost or undervalued [126], the cultural capital [127], and the authenticity and preserving of identity [73,128,129]. However, the degree of authenticity in the process of foregrounding tourism should be identified as "attraction-based identity" [130]. The discrepancies between the project and local residents is leading to the "deliberate" construction or adaptation of identity around the cultural experience [130] (p. 39), prioritising what the tourist at times perceived, i.e., banaliation, compared to what the heritage might truly be. A clear example of this is the FIM, which mixes

the local with the universal, is not centred on the participants and their experiences, and leaves the local population feeling detached from their roots (Int1). There is, too, the ongoing debate about the commodification of rural space [131] and culture [59,124], and the converting of authenticity, or identity, into merchandise, both of which call into question the development processes [132].

Since the 1990s, the issue of sustainability has been an additional construct in the debate over rural tourism [133] and cultural tourism [52,134,135]. In this regard, the most serious questions concern how to manage visitors to an area, how to control numbers, and how to establish limits [34], especially when there is a peak in demand for events, e.g., in FIM, or a marked seasonality, issues which require further study in order to establish reliable indicators of sustainability.

The institutional discourse, encapsulated in the document, "profitability, activation and sustainability, thinking of ways to generate wealth and further energize the local economy" [83] (p. 102) argues a contrary view to the capping of capacity, establishing a positive correlation between the number of visitors and development [136]. Nor does sustainability appear in the discourse of the business interests (Int5). Concern in this respect has only been voiced in the CAM, which fixes the maximum number of attendees at the FIM at 40,000 [76]. Indeed, there barely seems any awareness of the purely mathematical limits with respect to providing services, e.g., the town can accommodate 295 people/day and the catering service can provide 9104 meals with a turnover of eight services per seat, while the maximum capacity of the rest of the municipality is no larger than 307 people/day in terms of accommodation and ≥6000 meals.

It is evident that the rate of growth puts pressure on sustainability, as the ADPM noted in 2007 [82], and the effects of the town's inclusion on the LIPMP also need to be taken into account. The implications of potentially being declared a WHS could be manifold, and it is quite possible that they do not match expectations [43,137].

## 4. Conclusions

In the town of Mértola, a peripheral territory with fewer opportunities and a structural crisis, historical heritage, its conservation, and its value for tourism has converted into the comparative advantage [12] that generates opportunities [15,36] for local development [14]. However, the substantial amount of the heritage increases its conservation costs [38] and hinders the continuity of conservation projects.

The social, political, and institutional context [53] defines the processes of heritagisation and its value for tourism, which generate the dialectic between heritagisation and the exploitation or the commodification of heritage, as well as its overall perceptions [42] and the conceptions of development that are significantly dependent on dominant relationships and discourses [101]. Firstly, the cooperation [46–48] and then the competition between private (CAM, ADPM) and public (CMM) actors indicate contradictions and conflicts. The recovery of the collaborative approach [42] improves the results that are reflected by the increase of tourist supply and demand.

From the community perspective, the promotion of local participation in rural development policy-making processes has not been fulfilled. Diverse reasons and causes have progressively led to the social disaffection of the processes themselves, resulting in the loss of social and cultural capital, which is considered as the main asset of local development [10,11], and the overall control of the process [15]. Accordingly, there is an evident risk of conflicts as the consolidation of the external vision of the territory is opted for over the internal one [8].

As tourism enables diversification [12], it fosters the discourse of the heritage, tourism, and development correlation [12,18]. The milestones (FIM, LIPMP) have been fundamental for the development of supply and demand (quota). However, as shown by the data, the expectations generated by tourism have not been currently met [12,16]. Nonetheless, the favourable economic business trends, e.g., the generation of employment and income and increase in supply, as well as demographic trends, e.g., the slowing down of decline, can be observed. On the other hand, the adverse effects, such as the gentrification, overfrequentation, and marked seasonality, that compromise

sustainability have been identified [34,35]. Therefore, it is of crucial importance to determine and establish other activities to escape the theming caused by the specialisation.

In the context of the community participation [46,50], the improvements of the analysed processes are based on the comprehensive protection of the entire Mértola town, the development of planning processes, and a strategy that requires the participation of all stakeholders, public and private alike [49]. The establishment of limits through the perspective of carrying capacity and appropriate indicators will be necessary to achieve long-term sustainability [35]. Such planning becomes essential to avoid bottlenecks while aiming for an eventual declaration of the town as the World Heritage Site.

The limitations of the study mainly derived from the statistical series available, particularly after 1991, and the fact that primary information was not collected directly from the local population. On the other hand, the semi-structured interviews with the key stakeholders have provided extensive information and represented the pertaining views on the topic of this research.

Finally, the following research topics are proposed by the authors of this research, specifically, (a) in-depth studies of the relationships between economic, namely primary, secondary, and tertiary, activities and the impacts generated on them by the processes of heritage and tourism enhancement; (b) comparative studies of territories with similar characteristics to contextualise causes and consequences of the processes on local development; and c) analysis of issues related to employment, employability, gender issues, and quality of life of the local population in relation to heritage and its use in tourism processes. It is strongly considered that the future research on these themes could further contribute towards better understanding and more evidence-based analysis on the relation between tourism, heritagisation, and its values on sustainable local development.

**Author Contributions:** Conceptualization, F.J.G.-D., A.M.-P. and R.C.L.-G.; methodology, F.J.G.-D. and A.M.-P.; validation, F.J.G.-D., A.M.-P. and R.C.L.-G.; formal analysis, F.J.G.-D., A.M.-P. and R.C.L.-G.; investigation, F.J.G.-D., A.M.-P. and R.C.L.-G.; resources, F.J.G.-D. and A.M.-P.; data curation, F.J.G.-D.; writing—original draft preparation, F.J.G.-D., A.M.-P. and R.C.L.-G.; writing—review and editing, F.J.G.-D., A.M.-P. and R.C.L.-G.; supervision, R.C.L.-G.; funding acquisition, F.J.G.-D., A.M.-P. and R.C.L.-G. All authors have read and agreed to the published version of the manuscript.

**Funding:** This research received no external funding.

**Acknowledgments:** The authors are especially grateful to Susana Gómez Martínez, of the CAM, for her valuable assistance in arranging interviews and making available essential documentation.

**Conflicts of Interest:** The authors declare no conflict of interest.

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
