# Peer review of "Heritage, Tourism and Local Development in Peripheral Rural Spaces: Mértola (Baixo Alentejo, Portugal)"

_sustainability, doi:10.3390/su12219157_

Round 1

Reviewer 1 Report

Very interesting article about (not fulfilled) expectations of tourism in order to revive a region or to reverse a negative development trend. I read all the numbers, statistics and analyses with interest. However, I have the feeling that the picture drawn by the statistics does not seem to agree 100% with the statements at the beginning of the article. It has to be noted that this could also be due to the fact that most economic statistics cover a significantly shorter period than the reference period in the introduction (middle of 20th century). In this respect, the article would win if even older data were collected.

It would be helpful if there were references to the later statistics in the first half of the text so that one knows that they are coming. Otherwise too many questions remain unanswered at the beginning.

The authors wrote at the beginning that the population would no longer support the tourism strategies because they would not have produced the desired result (statement e) in the abstract). In my opinion, this is not treated accordingly within the discussion. The discussion is limited to the evaluation of the compiled statistical data. But the background to this reaction, which in my opinion is not supported by the numbers, would be very interesting.

In the same respect, the very interesting questions a) to f) could have been treated in more detail in becoming the leading points of the discussion.

Finally, the question of whether the authors have any ideas what could be done differently in Mértola ​​so that tourism brings the desired economic stimulation and how the population could be reunited behind the tourism concept. Both answers would be a great enrichment for the topic also for similar cases in other European regions.

Nevertheless, I did enjoy reading and learnt a lot about the area of Mértola and the local problems.

Author Response

  • Very interesting article about (not fulfilled) expectations of tourism in order to revive a region or to reverse a negative development trend. I read all the numbers, statistics and analyses with interest. However, I have the feeling that the picture drawn by the statistics does not seem to agree 100% with the statements at the beginning of the article. It has to be noted that this could also be due to the fact that most economic statistics cover a significantly shorter period than the reference period in the introduction (middle of 20th century). In this respect, the article would win if even older data were collected. Having resorted to official statistics from the National Institute of Statistics and the National Tourism Registry, it cannot be solved. The statistics and the text are reviewed so that they coincide 100%, making pertinent clarifications.
  • It would be helpful if there were references to the later statistics in the first half of the text so that one knows that they are coming. Otherwise too many questions remain unanswered at the beginning. We include references
  • The authors wrote at the beginning that the population would no longer support the tourism strategies because they would not have produced the desired result (statement e) in the abstract). In my opinion, this is not treated accordingly within the discussion. The discussion is limited to the evaluation of the compiled statistical data. But the background to this reaction, which in my opinion is not supported by the numbers, would be very interesting. The wording is improved and it is incorporated into the discussion (new section)
  • In the same respect, the very interesting questions a) to f) could have been treated in more detail in becoming the leading points of the discussion. We improve the wording and structures
  • Finally, the question of whether the authors have any ideas what could be done differently in Mértola so that tourism brings the desired economic stimulation and how the population could be reunited behind the tourism concept. Both answers would be a great enrichment for the topic also for similar cases in other European regions. They are included in proposals in the conclusion
  • We have incorporated the improvements discussed in the document.

Reviewer 2 Report

My comments are more than 600 words and are attached in a word document (Comments sustainability-967488). 

Author Response

  • The article is presented with a well-structured introduction, especially the second part, where the problem of the conflictive relationship between Heritage, Tourism, and local development is explained exhaustively. The research gap and the potential usefulness of the research are also adequately presented. At some point, the paragraphs are connected in a forced and non-linear way. But in general, it is striking and complete. The objectives of the study detailed at the end of the introduction do not agree with those explained in the abstract. This makes it difficult to understand the logic of the research. Revised
  • The materials and methods section are clear, but it is a little blunt. The methodology is well explained and referenced with respect to the procedure used for data collection. However, a more comprehensive explanation of the systematization of qualitative interview data is missing. The interviews are few and it is understandable that they wanted to extrapolate the most important information from each one and reflect it in the article, but a systematization of the information is desirable to highlight the contrast between the different opinions. We introduce an explanatory table, with the questions of the interview and the information provided
  • Section 3 on the case study is useful and relevant to place the reader in the context, however, I do not consider it is necessary to be in a separate section regarding the methodology: it could be included as " study context”. Both sections are united in one, divided into two epigraphs
  • It is in the results where one begins to detect little clarity in the distribution of information and connection with the proposed objectives. The three sections presented in the results (processes, conflictive perspective and effects and impacts) are too generic and do not provide the reader with the necessary information to understand their content. It is structured in results, discussion and conclusions, and the contents are specified
  • However, the presentation of the results is consistent with the proposed elements of analysis, which are: Roles and background of the stakeholders (partly present in section 4.1 and 4.2); the measures, instruments and actions implemented in the course of heritagization and implantation of cultural tourism (section 4.1); Critical assessment of the successes failures, results and overall impact (section 4.3). It is necessary to improve the titles of the sections or additionally also the distribution of the contents for better clarity. We rename the epigraphs
  • In addition, by linking the results section with the discussion one, the thread of analysis is lost. This may be due to the large number of results proposed. One proposal may be to separate the results sections from the discussion, or in any case it is necessary to deepen the discussion at the end of each results section, thus guiding the reader to the conclusions of the study. In addition, a further development of a discussion about the integration of the different results obtained from the elements of analysis would be appreciable. It is structured in results, discussion and conclusions, and the contents are specified
  • Another fundamental and missing section is conclusions. The article loses its usefulness by not summarizing the concrete findings obtained from the analysis. The lack of conclusions also reinforces the feeling of lack of discussion and argumentation of the results obtained from the study. Conclusions are added
  • If a concept is asserted with certainty, it must be adequately justified. For example, line 41-42, where a concept is expressed, which cannot necessarily be shared by all readers, cannot be introduced with "This is not to say". If it is decided to emphasize this concept because it is useful, it must be successively argued and referenced. New writing
  • Another example of the same issue is presented in line 49: the term "certainly" used in a concept that is not "obvious" requires argumentation. In general, in order not to generate controversy in the reader, it would be advisable to reduce the use of categorical terms. New writing
  • The connectors between sentences are not clear: e.g. line 50 "in this respect" is detached from the previous paragraph and the consequentiality is lost; New writing
  • line 43 "have emerged" implies a reasoning that is not well explained and difficult to understand. So, if the understanding is not immediate, it is appropriate to explain where the reflections have emerged from. Revised
  • In general, review the use of connectors between paragraphs and make sure that the concept referred to is unquestionable, and thus avoid confusion. New writing
  • In the paragraph that starts at line 89 there is too much use of contrast connectors that disorient the reading. It would be better to reorganize the paragraph and to make a cluster of all the aspects that help and those that do not help the development of tourism, so as to avoid chaotic writing and to obtain a more friendly style for understanding.  New writing
  • Section 3: Include on the map all the geographical references made in the text (lines 192, 204, 205 and 214) A new map is included
  • The acronyms used in the presentation of the results are a key point for interpretation. As they are a significant number and are used consistently in the writing, it would be appropriate to ensure that the process of interpretation of these is facilitated in some way. In the first quotation of acronyms, “hereinafter” is indicated
  • Add a graph/timeline to simplify understanding and reading. A timeline is included
  •  In addition, to facilitate subsequent discussion, some indicator could visually express the increased conservation and appreciation of the assets in relation to the events explained in the text. Images included

Round 2

Reviewer 2 Report

The authors have made most of the corrections. The article is now much clearer in terms of methodology and research context. Also, the results and the discussion have been deepened quite a bit creating a content of interest and usefulness. However, there are still three issues.

  • They have done a much better job of analyzing the data and distributing the information correctly among the sections. However, the difference between results and discussion remains very subtle and unclear. Now that the content is much better distributed and deepened, I suggest that the results and discussion sections be joined together but keeping the subtitles as they are now.

  • There is an error in the abstract. In lines 21 and 22, the phrase from the previous version should be eliminated: "the objectives of this study ..."

  • Map 1 on page 6 is unchanged. I still suggest that it would increase the clarity to include on the map all the geographical references made in the text.

Author Response

The authors have made most of the corrections. The article is now much clearer in terms of methodology and research context. Also, the results and the discussion have been deepened quite a bit creating a content of interest and usefulness. However, there are still three issues.

They have done a much better job of analyzing the data and distributing the information correctly among the sections. However, the difference between results and discussion remains very subtle and unclear. Now that the content is much better distributed and deepened, I suggest that the results and discussion sections be joined together but keeping the subtitles as they are now. Sections 3 and 4 are joined together (3. Result and discusion), 3.1 to 3.5 are listed.

There is an error in the abstract. In lines 21 and 22, the phrase from the previous version should be eliminated: "the objectives of this study ..." The error is eliminated (track changes).

Map 1 on page 6 is unchanged. I still suggest that it would increase the clarity to include on the map all the geographical references made in the text. The cartography made is entered (it was not entered in the change control).

Corrected (final) texts that were in red.

All changes are underlined in yellow: strikethrough (delete)
